# Synthesis and Vulcanization of Polymyrcene and Polyfarnesene Bio-Based Rubbers: Influence of the Chemical Structure over the Vulcanization Process and Mechanical Properties

**DOI:** 10.3390/polym14071406

**Published:** 2022-03-30

**Authors:** Arnulfo Banda-Villanueva, José Luis González-Zapata, Manuel Eduardo Martínez-Cartagena, Ilse Magaña, Teresa Córdova, Ricardo López, Luis Valencia, Sergio García Medina, Alejandro Medina Rodríguez, Florentino Soriano, Ramón Díaz de León

**Affiliations:** 1Research Center for Applied Chemistry, Blvd Enrique Reyna 140, San José de los Cerritos, Saltillo 25294, Mexico; abanda.d18@ciqa.edu.mx (A.B.-V.); gonzalezjl.d17@ciqa.edu.mx (J.L.G.-Z.); eduardocartaa@gmail.com (M.E.M.-C.); ilsma.rivera58@gmail.com (I.M.); trscordova@gmail.com (T.C.); ricardo.lopez@ciqa.edu.mx (R.L.); florentino.soriano@ciqa.edu.mx (F.S.); 2Biofiber Tech Sweden AB, Norrsken Hourse, Birger Jarlsgatan 57C, SE11356 Stockholm, Sweden; luisalex_val@hotmail.com; 3CIATEC, Omega 1201, Colonia Industrial Delta, Guanjuato 37545, Mexico; smedina@ciatec.mx (S.G.M.); lmedina@ciatec.mx (A.M.R.)

**Keywords:** bioelastomers, crosslinking, vulcanization, bio-based rubbers, bio-based monomers

## Abstract

The overuse of fossil-based resources to produce thermoplastic materials and rubbers is dramatically affecting the environment, reflected in its clearest way as global warming. As a way of reducing this, multiple efforts are being undertaken including the use of more sustainable alternatives, for instance, those of natural origin as the main feedstock alternative, therefore having a lower carbon footprint. Contributing to this goal, the synthesis of bio-based rubbers based on β-myrcene and *trans*-β-farnesene was addressed in this work. Polymyrcene (PM) and polyfarnesene (PF) were synthesized via coordination polymerization using a neodymium-based catalytic system, and their properties were compared to the conventional polybutadiene (PB) and polyisoprene (PI) also obtained via coordination polymerization. Moreover, different average molecular weights were also tested to elucidate the influence over the materials’ properties. The crosslinking of the rubbers was carried out via conventional and efficient vulcanization routes, comparing the final properties of the crosslinking network of bio-based PM and PF with the conventional fossil-based PB and PI. Though the mechanical properties of the crosslinked rubbers improved as a function of molecular weight, the chemical structure of PM and PF (with 2 and 3 unsaturated double bonds, respectively) produced a crosslinking network with lower mechanical properties than those obtained by PB and PI (with 1 unsaturated double bond). The current work contributes to the understanding of improvements (in terms of crosslinking parameters) that are required to produce competitive rubber with good sustainability/performance balance.

## 1. Introduction

More than 25 million tons of rubber are produced every year, most of which correspond to natural rubber (NR), or petroleum-based alternatives, such as butadiene-based rubbers alternatives (e.g., PB, NBR, and SBR) or isoprene-based rubbers (PI), among others [1]. There are several challenges from the current rubber alternatives. For instance, for NR, collection and processing are manual processes, and therefore, the production rate cannot satisfy the consumer demand (and the rubber industry is a growing market). On the other hand, due to the negative environmental impact of using petroleum as raw material, upcoming regulations are promoting the use of natural raw materials to produce bio-based materials. An ideal rubber material would show a good sustainability/performance balance, allowing to tailor the molecular weight characteristics and the copolymerization with other monomers for specific applications [2].

A prominent interest has emerged for the β-myrcene and *trans*-β-farnesene as sustainable alternatives, which are two bio-based monomers belonging to the family of terpenes. These monomers are obtained from natural sources that share similar (chemically and in terms of electronic properties) conjugated dienes such as 1,3-butadiene and isoprene [3,4,5]. In analogy to 1,3-butadiene and isoprene, the 1,3-diene moiety of β-myrcene and *trans*-β-farnesene enables its free-radical [6,7], anionic [8,9,10], and coordination [11,12] polymerization. Additionally, their (co)polymerization leads to sustainable rubbers [4,13] (PM and PF), which can be crosslinked via sulfur [3,4,10,14,15,16], thiol-ene [17], or other [18,19] types of crosslinking agents. Crosslinking of these bio-based rubbers is essential as all conjugated diene-based rubbers by themselves do not have the mechanical properties required for certain applications. Therefore, seen from an industrial point of view, the use of rubbers for engineering applications requires their transformation to a crosslinked state.

Several authors have studied the potential of terpene-based elastomers. For example, Zhang et al. prepared partially sustainable elastomers based on styrene and biogenic β-myrcene by living anionic polymerization and subsequently crosslinked them with sulfur. The presence of styrene in the copolymers improves the wet slip and rolling resistance properties [10]. On the other hand, Zhang et al. introduced β-myrcene into a polymerized solution of styrene-butadiene through anionic copolymerization to partially replace the butadiene in styrene-butadiene rubber. In addition, they carried out the crosslinking with sulfur. The results showed an improvement in the resistance to wet sliding with the incorporation of myrcene, in addition to an increase in tensile strength [16]. Although these reports support the interest in the synthesis of sustainable rubbers to generate materials, there are still very scarce scientific literature reports regarding the crosslinking and the evaluation of the mechanical properties of these bio-based terpenes. It is crucial to understand the optimum reaction conditions to produce performance-competitive bio-based rubbers, and benchmarking compared to readily commercial alternatives.

In this work, a neodymium-based (Ziegler–Natta type) catalytic system was used to synthesize the bioelastomers. This type of catalyst is highly active and leads to stereoregular polymers with high content of the 1,4-*cis* microstructure, which is needed for adequate elasticity, high fatigue and crack resistance, and low heat buildup in synthetic rubber. Vulcanized products obtain better mechanical properties (300% modulus, tensile strength, and elongation at breakage) with the content of the 1,4-*cis* microstructure [20]. The vulcanization of PM and PF was compared with PB and PI, using the same vulcanization process parameters and with/without carbon black, to elucidate the influence of the chemical structure on the crosslinking network and mechanical properties compared to conventional petroleum-based rubbers. Rubbers with variable molecular weights were also compared to obtain a wider panorama of their behavior during the vulcanization. Through this work, we aimed to set some guidelines toward the crosslinking of these bio-based rubbers and provide some insights into the performance of the resultant rubbers.

## 2. Materials and Methods

### 2.1. Materials

*β*-myrcene (>85% purity) was provided by Ventos (Barcelona, Spain) and *trans-β*-farnesene (≥95% purity) was supplied by Amirys (Emeryville, CA, USA). The 1,3-butadiene (>99%) and isoprene (99%) were provided by Sigma Aldrich (St. Louis, MO, USA). Prior to usage, *β*-myrcene and *trans*-*β*-farnesene were washed with a 2 M sodium hydroxide solution to remove the inhibitor, eventually dried with sodium sulfate salt, and distilled under reflux and sodium metal. Isoprene was distilled in the presence of metallic sodium before polymerization, and 1,3-butadiene was used as received. Neodymium versatate (NdV_3_) (NdV3-40, 0.54 M in hexane, reagent grade) was provided by Solvay (Brussels, Belgium); diisobutylaluminum hydride (Al(iBu)_2_H, 1 M in hexanes), dichlorodimethylsilane (Me_2_SiCl_2_, ≥98.5%), tetrahydrofuran (THF, 99%), deuterated chloroform (99.8% D) and cyclohexane (≥99%) was supplied by Sigma Aldrich (St. Louis, MO, USA); methanol (industrial grade) was provided by Quifersa (Saltillo, Coahuila, México); and pentaerythritol tetrakis (3-(3,5-diter-butyl-4-hydroxyphenyl)) propionate (Irganox 1010 commercial) was supplied by BASF (Ludwigshafen, Germany). Cyclohexane was washed with sulfuric acid to remove impurities and dried in a two-stage reflux distillation: the first with lithium aluminum hydride and the second with metallic sodium. Sulfur (industrial grade), dibenzothiazole disulfide (MBTS, industrial grade), zinc oxide (industrial grade), and stearic acid (industrial grade) were purchased from Specialty Supply (Monterrey, Nuevo, México). Carbon black N330 (industrial grade) was provided by Continex (Houston, TX, USA).

### 2.2. Methods

#### 2.2.1. Synthesis of Elastomers

The synthesis of the rubbers (both biobased and non-biobased) was carried out using a ternary catalytic system consisting of Al(iBu)_2_H, NdV_3_, and Me_2_SiCl_2_. The polymerization was carried out until reaching conversions close to 100%, in a 1 L stainless-steel reactor under a nitrogen atmosphere. The preparation and aging of the catalytic system were carried out in a glove box under argon atmosphere. First, the cyclohexane and monomer were added into the steel reactor, and it was sealed and heated until reaching 70 °C. Then, the catalytic system was injected into the reactor to initiate the polymerization reaction. The polymerization was terminated with acidified methanol and an antioxidant (Irganox 1010) was added. The resultant rubber was washed with methanol and vacuum-dried until constant weight.

#### 2.2.2. Formulation and Crosslinking

The different formulations were mixed in a two-roll mill Schwabenthan Polymix 40T (Maschienenfabric Fr. Schwabenthan & Co. Kg., Berlin, Germany) at a temperature of 50–60 °C with a nip gap of 0.5 mm. The rubber compounding was started by mastication of the rubber on the two-roll mill; then, the other ingredients were added until a homogenous compound was obtained in the following order: zinc oxide (enhances the effect of accelerators), stearic acid (as a softener and a filler-dispersing agent), sulfur (crosslinking agent), and MBTS (increases the speed of the process). Scorch time (t_s2_), optimum cure time (t_c90_), minimum torque (M_L_), and maximum torque (M_H_) were determined from the cure curve by a rubber process analyzer (RPA Elite) (TA Instruments, New Castle, DE, USA) (crosslinking parameters are described in the characterization section). The samples were crosslinked to 160 °C for the conventional (CV) and efficient (EV) vulcanization, at t_c90_ on a Carver auto four/30H-12 automatic hydraulic laboratory press (Wabash, IN, USA). The different formulations are shown in Table 1. Each system was also reinforced with carbon black (CB) to represent the compounds used in the manufacture of tires.

### 2.3. Characterization

#### 2.3.1. Synthetized Polydienes

The microstructure of the synthesized rubbers was determined via ^1^H and ^13^C Nuclear Magnetic Resonance (NMR) spectra in a 400 MHz Bruker Advance III NMR spectrometer (Billerica, MA, USA). The macrostructure of synthetized rubbers, molecular weights, and their distributions were calculated with an Agilent Gel Permeation Chromatograph (GPC) model PL-GPC50 (Santa Clara, CA, USA), equipped with a refractive index detector, at a flow rate of 1 mL/min of THF at 40 °C, calibrated using polystyrene standards.

#### 2.3.2. Crosslinked Rubbers

The crosslinking parameters were obtained through an oscillating disc rheometer (ODR) using a rubber process analyzer from TA Instruments (RPA elite). A high-resolution solid-state Nuclear Magnetic Resonance ^13^C CP-MAS study was carried out in 500 MHz Bruker Advance III equipment to determine the crosslinking bonds. The mechanical properties were determined using an MTS criterion model 43 (MTS System Corporation, Eden Praire, MN, USA), and tensile strength tests were performed at room temperature at a speed of 500 mm/min according to ASTM D412. The crosslinking density was determined by measuring the swelling balance by placing the previously crosslinked rubbers in toluene for 72 h at 30 °C. At the end of this period, the samples were weighed by removing the excess of toluene with paper and later placed in a vacuum oven at 60 °C to remove the solvent trapped within the sample and weighed again to constant weight. Afterward, the volume of the rubber fraction ($V_{r}$) was calculated with the following equation [3,21]:(1)$$V_{r}=\frac{\left(W_{i}-FW_{m}\right)/\rho_{r}}{\left(W_{i}-FW_{m}\right)/\rho_{r}+W_{s}/\rho_{s}}$$
where $W_{i}$ is the weight of the dry sample after swelling, *F* is the fraction of the insoluble weight of the sample, $W_{m}$ is the weight of the sample before swelling, $W_{s}$ is the weight of the solvent absorbed by the sample, $\rho_{s}$ is the density of the solvent (0.867 g/cm^3^), and $\rho_{r}$ is the density of rubber. The density of rubber was determined using the hydrostatic weighing method, where the apparent weight of the sample is measured in two different media, in water ($W_{w}$) and air ($W_{a}$), applying the equation [22]:(2)$$\rho_{r}=\frac{W_{a}\rho_{a}-W_{m}\rho_{w}}{W_{w}-W_{a}}$$
where $\rho_{w}$ (0.9977 g/cm^3^ at 22 °C) and $\rho_{a}$ (0.0012 g/cm^3^) represent the density of water and air, respectively.

The crosslinking density (*v*) was calculated with the Flory–Rehner equation [3,21]:(3)$$-\left[ln\left(1-V_{r}\right)+V_{r}+\chi V_{r}^{2}\right]=\frac{\rho_{r}}{M_{c}}V_{s}\left(V_{r}^{\frac{1}{3}}-\frac{V_{r}}{2}\right)$$
(4)$$=\frac{1}{2M_{c}}=-\frac{\left[In\;\left(1-V_{r}\right)+V_{r}+\chi V_{r}^{2}\right]}{2\rho_{r}V_{s}\left(V_{r}^{1/3}-\frac{V_{r}}{2}\right)}$$
where $M_{c}$ is the molecular weight between the crosslinks, $V_{s}$ is the molar volume of the solvent (106.29 mL/mol), and *χ* is the solvent–rubber interaction parameter; this was determined with the Hildebrand solubility parameters with Equation (5) [3,21]:(5)$$\chi =\chi_{\beta }+\frac{V_{r}}{RT}{\left(\delta_{s}-\delta_{r}\right)}^{2}$$
where $\delta_{s}$ (8.97 cal/cm^3^) and $\delta_{r}$ are the solubility parameters of the solvent and rubber, respectively, *R* is the ideal gas constant (1.987 cal/Kmol), *T* is the absolute temperature (K), and $\chi_{\beta }$ is the entropic contribution (typically 0.34) [23,24]. The $\delta_{r}$ was determined based on the chemical structure of the rubber, using the molar group attraction constants (*G*) with Equation (6) [25]:(6)$$\delta_{r}=\frac{\rho_{r}\sum^{\;}G}{M}$$

Hardness tests were made based on ASTM D2240 with a type A durometer. The compression set test was performed based on ASTM D395, Method B for 22 h at 70 °C. The abrasion resistance test was carried out under the ISO-4649:2017, method A. Dynamic Mechanical Analysis (DMA) was carried out in tension mode at 3 °C/min with a frequency of 1 Hz and strain of 10 µm in DMA Q800 equipment (TA Instruments, New Castle, DE, USA). The temperature range was −120 to 20 °C for PB and −90 to 20 °C for PI, PM, and PF.

## 3. Results and Discussion

### 3.1. Synthesis of Rubbers

The synthesis of PB, PI, PM, and PF was carried out via coordination polymerization using a ternary neodymium-based catalytic system comprising NdV_3_/Al(iBu)_2_H/Me_2_SiCl_2_. This catalytic system is known to polymerize dienes with high *cis* microstructure control [26]. NdV_3_ is the catalyst, while Al(iBu)_2_H is an alkylaluminium acting as a co-catalyst acting as Lewis’s acid abstracting alkyl groups from the Nd molecule, thus creating active sites, and Me_2_SiCl_2_ is a halide donor that promotes catalytic activity and stereocontrol [27,28].

The macro- and microstructural properties of the rubbers (molecular weight characteristics, and *cis* microstructure) are shown in Table 2. Two different molecular weights of rubber were used for each material for subsequent vulcanization, denoted as H (high molecular weight) and L (low molecular weight). This was performed to investigate the influence of molecular weight on the final characteristics of the crosslinked network. All the synthesized rubbers exhibited a high *cis* content (>95%), characteristic of the employed catalytic system, and molecular weights ranging from 292 to 1161 kg/mol (see Table 2). The chemical structure of the monomers and the resultant high-*cis* rubbers are shown in Figure 1.

### 3.2. Vulcanization of Rubbers

The crosslinking of the resultant rubbers was performed via sulfur vulcanization. The sulfur-crosslinked systems are described in the literature in three types: CV, EV, and semi-efficient vulcanization (semiEV). The difference among them lies in the accelerator and sulfur ratio (A/S), being 0.1–0.6 for CV, 2.5–12 for EV, and an intermediate range for semi-EV. The final properties of the resultant rubbers depend on the amount of mono- (C-S-C), di- (C-S_2_-C), and polysulfides (C-S_x_-C) bonds that crosslink the polymer chains. The network formed by the CV is composed mostly of di- and polysulfides (95%) bonds, while the EV is predominantly integrated with monosulfide (~80%) bonds, and the semi-EV is a mixture balanced by both [24,29,30,31,32]. For the rubbers synthesized here, both CV and EV were used for the vulcanization process.

The crosslinking reactions were analyzed by measuring the torque (dNm) as a function of time, obtained from the cure curves at a temperature of 160 °C. The results are shown in Figure 2 and Figure 3. Values of torque and other data obtained from cure curves are shown in Appendix A of the supporting information. The influence of the rubber molecular weight was evident in the unreinforced and CB-reinforced rubbers, showing a higher torque at high molecular weights. Increasing the molecular weight and the linearity of the polymer chain structure (such as PB synthesized with Ziegler–Natta catalysts based on neodymium) increases the stiffness of the rubber, whereas when the rubber has a low molecular weight and a structure with long branches, the material exhibits lower stiffness [33]. Therefore, PB rubber showed the highest torque in both unreinforced and CB-reinforced compounds. This behavior was shown in the rubbers with either high (Figure 2a and Figure 3a) or low molecular weight (Figure 2b and Figure 3b). In general, the trend in the torque values for the synthesized rubbers is PB > PI > PM > PF for high- and low-molecular-weight rubbers, and following the same behavior if they are CB-reinforced. PF showed the lowest torque values, thus less rigidity, especially at the low molecular weight at which they were unable to be measured, due to high fluidity (thus not shown in Figure 2b and Figure 3b).

Higher torque was observed when using EV, suggesting a matrix mainly crosslinked via monosulfide bonds, limiting the network molecular mobility, and reducing the flexibility, as it has been previously described in the literature when compared to a flexible crosslinked network made up of a higher proportion of polysulfide bonds using CV [29,30,32]. In the rheometric curves, a slight reversion is observed in the rheometric curves of the rubbers with the EV system. In the literature, this phenomenon has been presented in this type of system due to the conversion of disulfide to monosulfide crosslinks [30,34]; however, in our case, this could be due to the absence of antioxidant [35].

On the other hand, mixing CB with PB and PI required more work due to the rigidness of the material, which was not presented when adding it to PM and PF, making it easier to the mixing process. According to Appendix A, CB had a positive impact on the t_s2_ and t_c90_, both decreasing in most cases. This is attributed to the reinforcing effect of CB, which restricts the mobility of the chains so that a greater amount of energy is accumulated (heat; this is discussed in greater detail in the section corresponding to the DMA) as it is not dissipated by molecular movement, promoting the breaking of the bonds in the accelerant and sulfur. The incorporation of the CB increases the stiffness of the material presumably due to electrostatic attractions (physical bonds) that increases the rigidity and reduce the flexibility of the crosslinking network.

### 3.3. Crosslinking for Each Vulcanization System

The bonds formed at the crosslinking network of PB and PI have been studied in several works, describing a variety of structures formed by mono-, di-, and polysulfides bonds, as well as cyclization, carbon-carbon crosslinks, and accelerator residues, as shown in Figure 4 [36,37]. Figure 5 and Figure 6 show the crosslinking bonds that can be formed in *cis*-PI and *cis*-PB, respectively [38]. The possible crosslinking network for PM and PF has not been extensively studied as in the case of PB and PI. Therefore, it is important to elucidate the influence of the chemical structure (showing a branch with one or two isoprenic units, respectively, see Figure 7) of the polyterpenes over the rubber vulcanization.

However, the unsaturated branches present in the PM and PF may imply a greater availability for the formation of crosslinking bonds, which are not present in PB and PI. Various assumptions arise to understand the behavior of the crosslinking network in the PM and PF. Looking at the van der Waals forces and entanglement molecular weight (Me), it is possible to find a relationship with the stiffness and toughness of the rubbers. PB, being a linear polymer, has a greater number of electrostatic attractions between the atoms of the rubber chains, producing a material with poor fluidity and high melt viscosity by a low Me (1.8 kg mol^−1^). In contrast, the van der Waals forces and melt viscosity decrease with the size of the branching present in the main chain of PI (Me = 5.4 kg mol^−1^), PM (Me = 18 kg mol^−1^), and PF (Me = 50 kg mol^−1^), where PM and PF present more fluidity and therefore less rigidity [37,38,39,40].

On the other hand, if we suppose that PI presents *n* possibilities for the formation of the crosslinking network by presenting a single unsaturation, PM will have *2n* possibilities when presenting two unsaturations, and, finally, the PF will have *3n* possibilities when presenting three unsaturations. This extra number of possibilities gives PM and PF a better distribution of the crosslinking sites, which directly impacts the crosslinking network. That is, if we consider that the chemistry of the reaction is proportional to these possibilities, PM then needs a double amount of crosslinking agent to equal the properties obtained by PB and PI, while PF should require a triple amount of crosslinking agents to obtain the same properties. For this reason, it is necessary to carry out a series of experiments to confirm that the increase in the amount of crosslinking agents has positive effects on the properties of the crosslinking network. However, the literature reports that an increase in the amount of sulfur, or accelerator, has a negative effect for PB and PI, by reducing the mechanical properties [26].

To validate the possible configuration of the crosslinking network, 13C CP-MAS solid-state NMR of the aliphatic region (10 to 75 ppm) was performed for the PI-H, PM-H, and PF-H samples crosslinked with the CV system, which is shown in Figure 8. Analyzing the NMR spectra corresponding to the PI-H compound, we observe that at least four structures were determined for the formation of the crosslinking network, with chemical shifts at 53.45, 51.32, 45.40, and 42.26 ppm.

For PM-H, it was only possible to determine one crosslinking structure that fits the spectrum (38.95, 35.81, 34.06, 28.06, 25.85, 18.33, and 15.33 ppm), corresponding to crosslinking through branches by monosulfide bonds and cycle formation, showing a better distribution of the sulfur by reducing the presence of di- and polysulfide bonds (although the formation of other structures formed by di- and polysulfide bonds is not ruled out). It is unlikely that crosslinking occurs in the main chain due to low chemical shifts in comparison with the PI-H (signals less than 45.26 ppm). Finally, for the PF-H rubber (42.68, 39.38, 37.95, 31.66, and 20.83 ppm), as it shows a chemical structure similar to PM-H, the formation of crosslinking bonds with the main chain seems to be also unlikely. Instead, crosslinking through the branches and cycles formation are also suggested.

The low-field signal at 42.68 ppm describes the presence of disulfides, polysulfides, and even crosslinking with the main chain, but the overlapping of signals does not allow the determination of what is the dominant crosslinking mechanism. The EV system theoretically presents the same structures proposed for the CV system, but mostly made up of monosulfide bonds. This shows that the crosslinking network is more open in proportion to the size of the branch, and it has a negative influence on the mechanical properties for PM and PF.

### 3.4. Crosslinking Density and Mechanical Properties of Rubbers

The mechanical properties of a vulcanized rubber also strongly depend on the crosslinking density. According to Table 3 and Table 4, where mechanical properties and crosslinking density are reported, respectively, generally higher values for tensile strength and elongation at break are observed in formulations with high-molecular-weight rubbers. This condition suggests that the high molecular weights offer a more effective crosslinking matrix because the chains tend to become entangled with each other, thus generating a more rigid matrix and requiring a greater force to break the entanglements and the linkage bonds of the network [40]. In contrast, low-molecular-weight rubbers tend to entangle less, and even short chains could be considered to function as a lubricant for longer chains. On the other hand, considering the above and visualizing the properties presented by the crosslinking network of PB, it could be assumed that it would present the best results for elongation at break. However, it maintains this parameter in a range of 662–825%, surpassed by PI with a greater elongation at break of 987–1262% (some exemplary stress-strain curves are shown in Appendix A of the supporting information). This characteristic has already been studied by several authors who consider that the PI crosslinking network tends to form crystalline regions that are favored when the material is stretched and are not observed under other conditions [41,42,43,44,45]. This hypothesis guided the analysis of PM and PF, hoping that this condition would be replicated by the crosslinking network. However, such formulations exhibited lower elongations at break for PF and PM contrasting with PI and PB, indicating that these rubbers do not present any type of crystallization, determining that the chemical structure of PM and PF reduces the effectiveness of the crosslinking network in proportional ways to the size of the branch.

On the other hand, the presence of CB reinforced the rubbers in terms of tensile strength and reducing the elongation at break of PB and PI. This is due to a reduction in the percentage of rubber when incorporating CB, agglomerates, and the increase in crosslinking density that makes a more rigid material. Furthermore, the presence of CB minimizes the stretch-induced crystallization in PI rubbers [46]. However, in PM compounds, there was a lower reduction in elongation at break than PB and PI rubbers, suggesting that PM allows the incorporation of CB without significantly affecting the tensile strength, and it is more compatible than PB and PI, possibly because the crosslinking network is more open than the PB and PI rubbers. This is corroborated with the increase in tensile strength and elongation at break values for the case of reinforced PF rubbers.

The moduli at 100%, 200%, and 300% elongations are related to the crosslinking density. When this property increases, the moduli value reached is greater (for both vulcanization systems) and there is a decrease in the elongation at break due to a greater restriction of the crosslink joints on the mobility of intercrosslink chains [47].

The hardness and compression set of the crosslinked rubbers are shown in Table 3 and Table 4. It is generally observed that most materials crosslinked with the CV system have a higher hardness for high-molecular-weight rubbers, which is closely related to the increase in the crosslinking density because of the restriction of the crosslinking linkages on the mobility of the chains [47,48]. In the crosslinking rubbers with the EV system, the PM and PF show in the same way a higher hardness with the rubbers of higher molecular weight; however, this condition was not presented for PB and PI. The unreinforced compounds of PB with CV and EV had the highest hardness of the evaluated materials, suggesting that it is an intrinsic property of PB produced by its chemical structure and the formation of a more closed crosslinking network (linear chain). However, for reinforced and unreinforced rubbers, a clear trend was not found in relation to molecular weight, but a reduction in hardness of reinforced PI, PM, and PF rubbers was observed and evaluated in CV and EV. This proposes that the hardness depends on the chemical structure of PI, PM, and PF, which results in a more open crosslinking network (given by their branches), therefore being more flexible and with less hardness. In the case of CB-reinforced materials, all rubbers showed an increase in hardness. According to the literature, this is due to the influence of CB on the acceleration of the crosslinking kinetics, and the changes induced by CB in the crosslinking density, which is increased in all reinforced rubbers. The increase in hardness leads to a material with greater elasticity and less flexibility [49].

The compression set test was carried out to determine the ability of rubber to retain its elastic properties after prolonged compression loads at a certain temperature. The analysis of this parameter is essential to determine when the rubber material is suitable to be applied as parts of dampers and seals, which are subject to compression loads. A small compression set value refers to the ability of the sample to maintain original thickness and low damping properties, while a large value indicates lower stiffness with broader damping properties. It is also known that this parameter is another property that can be used to determine the degree of elasticity [50].

All the materials without reinforcement exhibited a lower compression set value at a higher molecular weight of the rubber, and according to the literature, this indicates that the presence of strong chemical bonds in the crosslinking network gives a better ability to recover the initial shape and undergo a less permanent deformation [24,51]. The PB compounds showed lower compressions set with the CV system of 33% and 62%, while the other compounds were deformed from 83% to 99%. On the other hand, the compounds evaluated with the EV system presented better resistance to permanent deformation in this order: PM-H > PB-H > PF-H > PB-L, and the other rubbers had percentages >98%. It should be noted almost all materials have high compression set values; therefore, they suffer a high permanent deformation. Furthermore, by adding CB as reinforcement in general, the compression set is reduced as it gives the material a certain rigidity to withstand the compression load for both CV and EV systems used, even being higher for compounds evaluated with EV.

Figure 9 shows the abrasion resistance of CB-reinforced and -unreinforced composites at high molecular weight evaluated in EV. It is observed that the compounds PI-H, PM-H, and PF-H do not appear in Figure 9, because the materials were soft, and it was not possible to determine this characteristic. However, from this group of compounds, the PB-H could be assessed, showing an abrasion resistance of 182 mm^3^, which, in turn, was reduced to 40 mm^3^ with the incorporation of CB, thereby losing a smaller amount of material. The reinforced compounds obtained resistance to abrasion in the following order: PB-H-CB > PI-H-CB > PM-H-CB > PF-H-CB, with this information again relating to the effect of the chemical structure of the evaluated rubbers, corroborating that the linear chains of PB form a closed crosslinked network that limits the detachment of particles by abrasion. Instead, the branched chains in PI, PM, and PF constitute an open crosslinked network that facilitates the loss of material by abrasion as follows: PF-H-CB > PM-H-CB > PI-H-CB.

### 3.5. Dynamic Mechanical Analysis (DMA) of Crosslinked Materials

In order to determine the viscoelastic properties of high-molecular-weight rubbers, an analysis using Dynamic Mechanical Analysis (DMA) was carried out; some values obtained are shown in Table 5, such as storage (E’) and loss modulus (E”) at −40 °C. For the samples unreinforced and vulcanized with the EV system, the storage modulus values reached are much higher compared to the CV system; this is attributed to the type of monosulfide bonds produced during crosslinking, as mentioned in Section 3.2. In addition, it is observed that the storage modulus is higher for the PI-H (EV system) compound, indicating that the mobility of the crosslinking network in the vitreous region is more limited than in the other compounds.

The loss factor (or damping) factor, also called Tan δ, which is the ratio E”/E’, is an indicator of the dynamic behavior that occurs in the glass transition region and is related to the mobility of the polymer chains, as well as the amount of energy dissipated by the material [52]. The damping factor of PF and PM is as remarkably high as PI, indicating better interaction between polymer chains in the crosslinking network [52]. In addition, this is related to the length of the branch of the lateral alkene groups that would lead to a greater loss of energy under dynamic load [10].

The effect of carbon black as a reinforcer caused an increase in the storage modulus in all compounds, as can be seen in the Figure 10, but more markedly in the PI-H-CB (EV) and PF-H-CB (EV), indicating that these compounds maintain an excellent molecular interaction with carbon black [52,53]. All compounds show a drop in storage modulus as they approach the glass transition temperature (T_g_) until they exceed it and enter in the rubbery region where the long-range coordinated movement of the chains makes the elastic modulus unable to be detected and reduces to zero [54]. The information obtained by the loss modulus is proportional to the storage modulus, thereby verifying the viscoelastic behavior of the material, which is characteristic of any elastomer.

The materials vulcanized with the CV and EV system (Figure 10) with reinforcement present lower values of Tan δ and broader curves, indicating a reduction in the heat build-up and damping capability [10], which is because CB acts as a barrier to the mobility of the rubber chains, so they accumulate more energy (heat). The temperature of the Tan δ peak is related to the glass transition temperature (T_g_). In this work, the incorporation of CB does not seem to overly displace or affect the T_g_, due the value of this parameter showing minimal changes in the samples with and without reinforcement.

## 4. Conclusions

In this work, we compared the properties of terpene-based bio-rubbers to conventional fossil-based rubbers (polyisoprene and polybutadiene), in order to set some guidelines toward the crosslinking of these materials, and thus the production of products with competitive performance. Using the herein processing conditions, polymyrcene and polyfarnesene produce materials with lower mechanical properties than those obtained by PB and PI, presumably because they form crosslinking networks mainly with the branches according to the RMN characterization, making more open networks that decrease the effectiveness of the crosslinked material. The crosslinking with branches and the absence of induction crystallization compared with PB and PI cause a decrease in some properties such as tensile strength. 

Generally, the incorporation of carbon black generally improves the mechanical properties of rubber compounds and reduces the crosslinking time, elongation at break, and compression set, while the hardness and resistance to abrasion increase. However, PF compounds show an increase in elongation at break upon CB addition, indicative of better network interaction with the additive. Unlike unreinforced compounds with PI, PM, and PF, which dissipate energy better than compounds with PB, the addition of CB reduces the energy dissipation, thereby accumulating heat. It appears that the mechanical properties such as tensile strength, elongation at break, and compression set of crosslinked rubbers depend on molecular weight, and these can be improved by increasing the molecular weight, while the hardness and resistance to abrasion depend on the chemical structure, as, in linear rubbers, these properties improve, and in branched rubbers, they decrease as a function of the size of the branch. This work provides guidelines for future work, which can take the herein developed knowledge to optimize formulations that can compete with the current fossil-based rubbers.

## Figures and Tables

**Figure 1 polymers-14-01406-f001:**
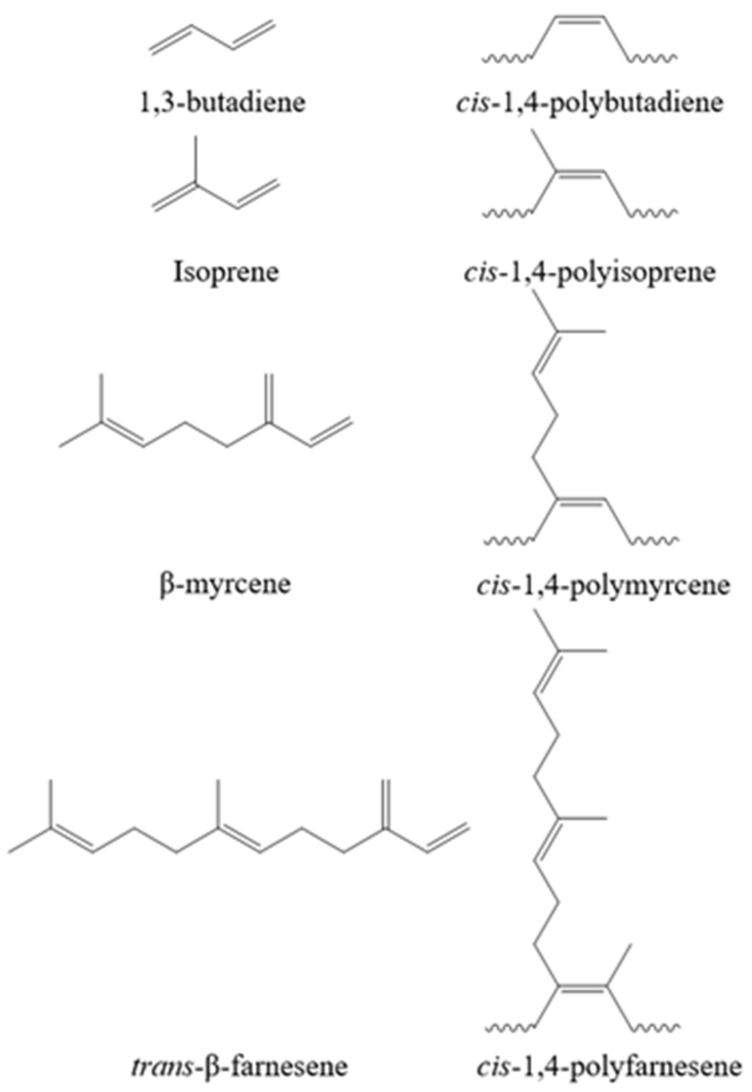
Chemical structure of monomers and the resultant high *cis* rubbers.

**Figure 2 polymers-14-01406-f002:**
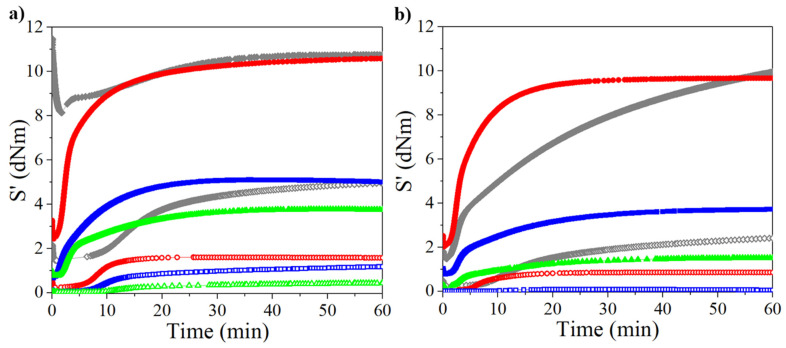
Rheometric curves for crosslinked rubbers through CV at 160 °C for high- (**a**) and low-molecular-weight (**b**) rubbers (open and filled symbols represent unreinforced and reinforced rubbers, respectively). Legend: diamonds→PB, circles→PI, squares→PM, triangles→PF.

**Figure 3 polymers-14-01406-f003:**
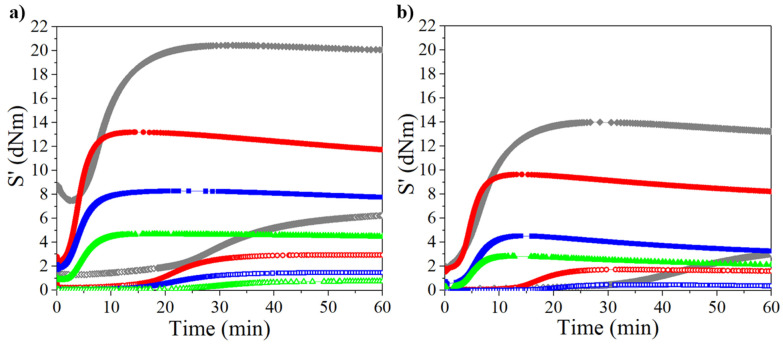
Rheometric curves for crosslinked rubbers through EV at 160 °C for high- (**a**) and low-molecular-weight (**b**) rubbers (open and filled symbols represent unreinforced and reinforced rubbers, respectively). Legend: diamonds→PB, circles→PI, squares→PM, triangles→PF.

**Figure 4 polymers-14-01406-f004:**
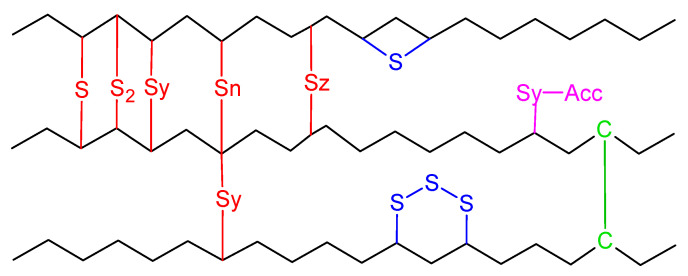
Structures formed as a result of intermolecular curing [37]. Adapted with permission from Ref. [37]. Copyright (2022) American Chemical Society.

**Figure 5 polymers-14-01406-f005:**
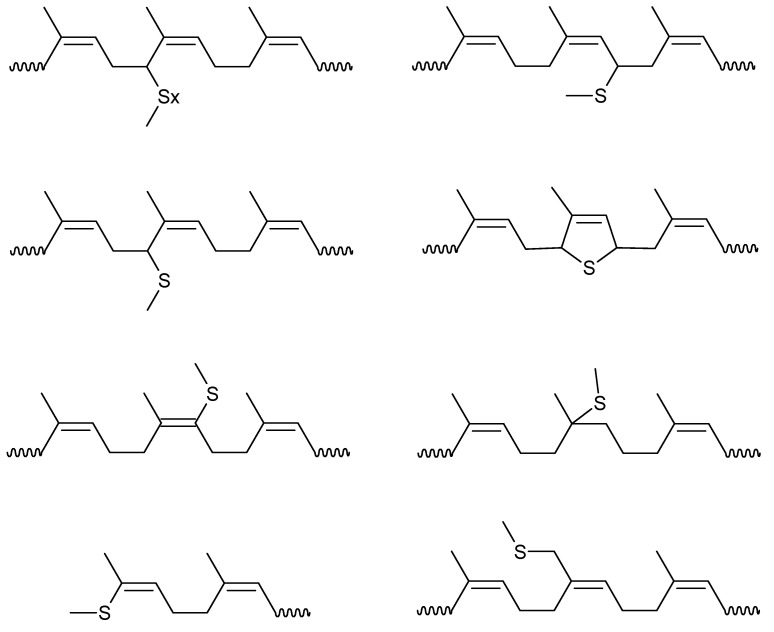
Proposed structures that are produced through the vulcanization of *cis*-PI [38]. Adapted with permission from Ref. [38]. Copyright (2022) Elsevier Inc.

**Figure 6 polymers-14-01406-f006:**
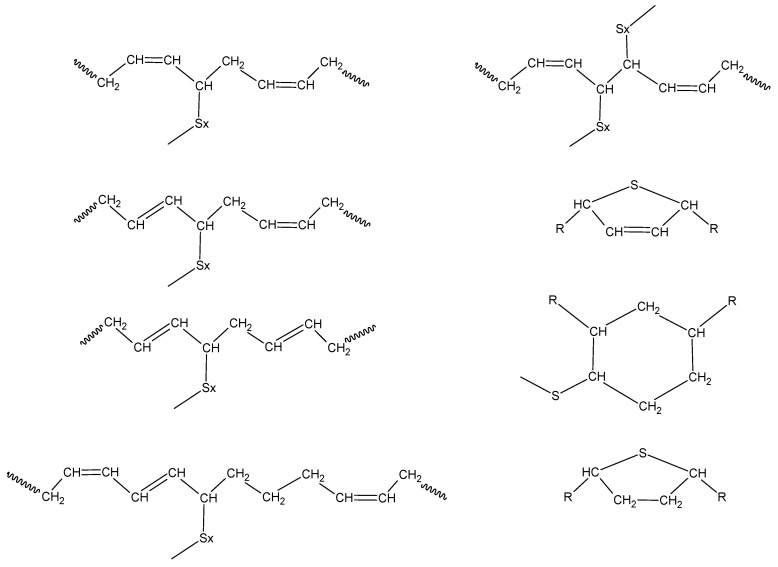
Proposed structures that are produced through the vulcanization of *cis*-PB [38]. Adapted with permission from Ref. [38]. Copyright (2022) Elsevier Inc.

**Figure 7 polymers-14-01406-f007:**
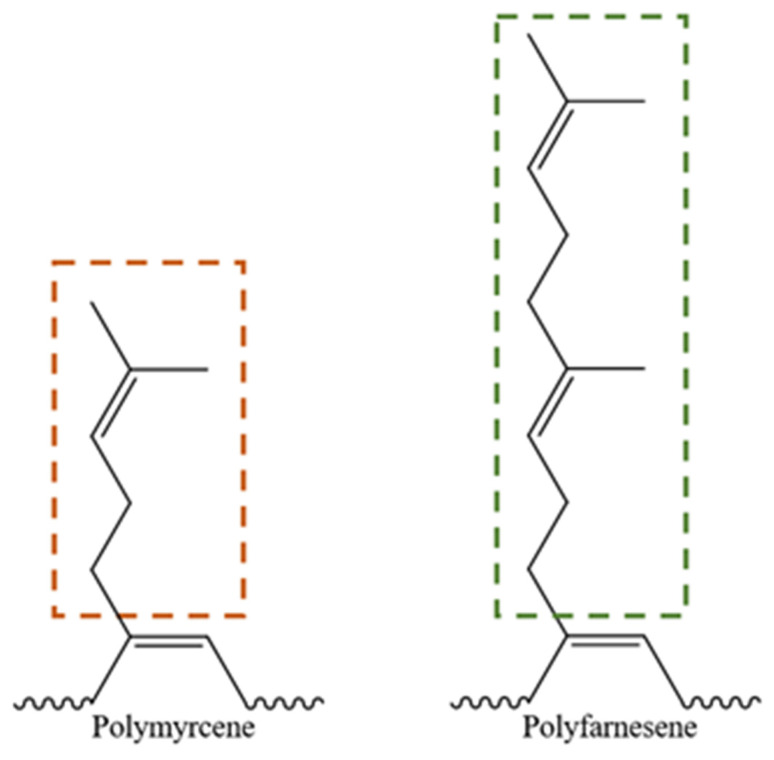
Branch structure of *cis*-PM and *cis*-PF.

**Figure 8 polymers-14-01406-f008:**
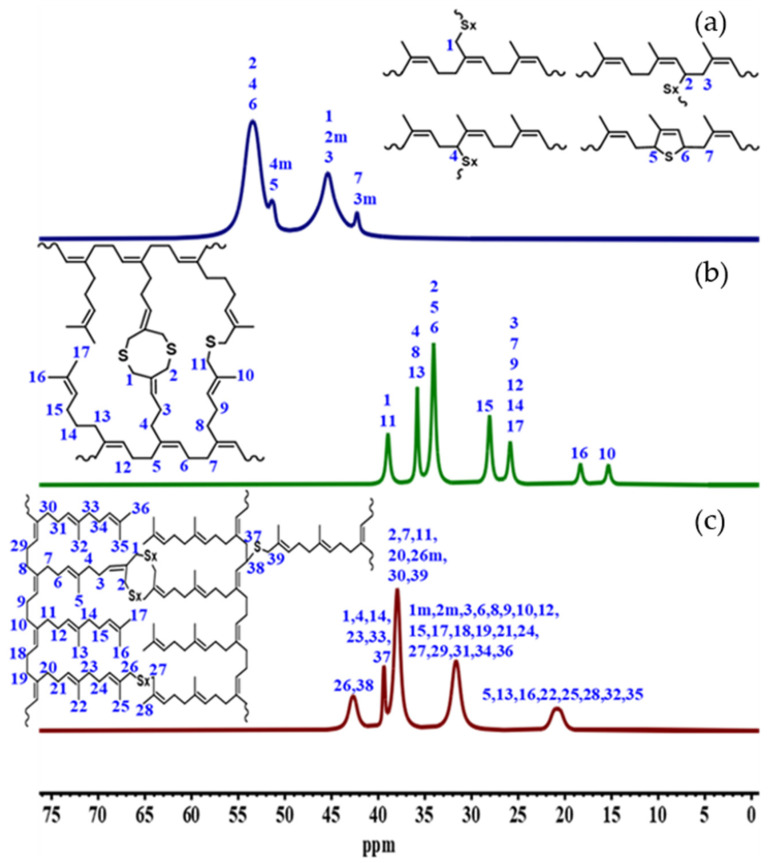
^13^C CP-MAS solid-state NMR spectrum of crosslinked (**a**) PI-H, (**b**) PM-H, and (**c**) PF-H using the CV system (the spectra have been treated with deconvolution and noise removal).

**Figure 9 polymers-14-01406-f009:**
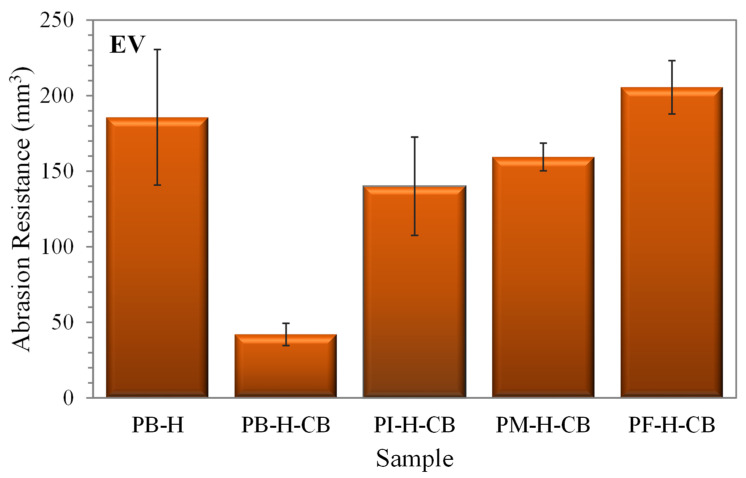
Abrasion resistance of reinforced and unreinforced compounds evaluated with EV system.

**Figure 10 polymers-14-01406-f010:**
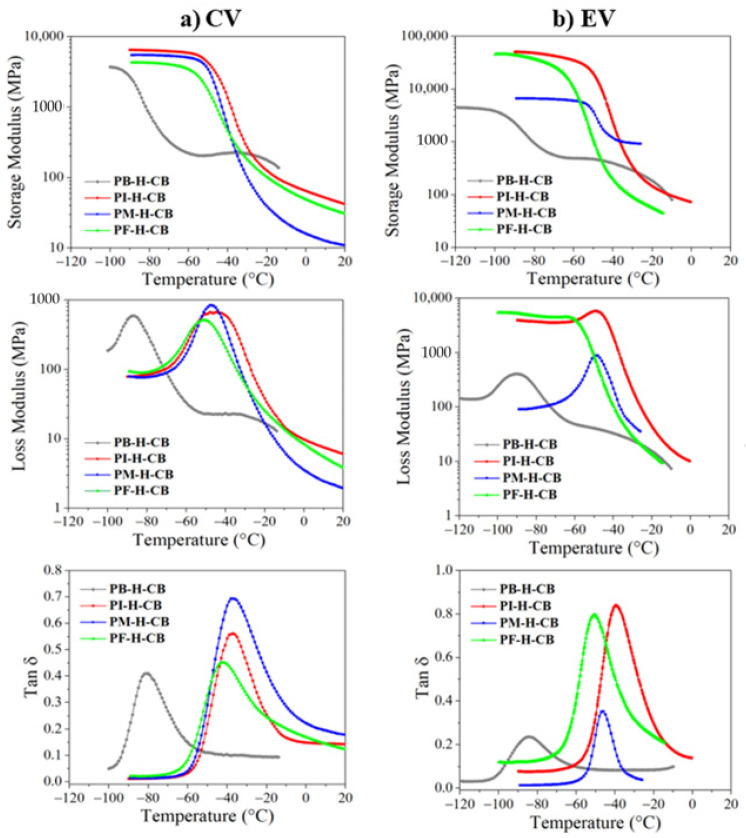
Storage modulus, loss modulus, and Tan δ of the crosslinked reinforced materials through the system CV (**a**) and EV (**b**) by DMA.

**Table 1 polymers-14-01406-t001:** Formulations of CV and EV.

Component	Content (phr ^a^)			
	CV		EV	
	WR ^b^	CB ^c^	WR ^b^	CB ^c^
Rubber	100	100	100	100
Zinc oxide	5	5	5	5
Stearic acid	2	2	2	2
Carbon black N330	-	50	-	50
Sulfur	2	2	0.5	0.5
MBTS accelerator	0.5	0.5	2	2

^a^ phr = per hundred parts of rubber, ^b^ without reinforcement, ^c^ with CB as reinforcement.

**Table 2 polymers-14-01406-t002:** Macro- and microstructural properties of synthesized rubbers.

Rubber	[Monomer]: [Nd]	*M_w_* (Kg/mol)	Ð *^a^*	*Cis* (%) *^b^*
PB-H	12,600	693	3.76	99.20
PB-L	6300	302	2.91	96.04
PI-H	7900	865	4.11	95.08
PI-L	3950	292	2.46	94.46
PM-H	2000	791	3.80	95.83
PM-L	1000	332	3.54	95.78
PF-H	780	1161	3.98	95.56
PF-L	600	456	4.92	95.42

*^a^* Determined by GPC using polystyrene standards; *^b^* calculated from the ^1^H and ^13^C NMR. All reactions were carried out at 70 °C pressurized with nitrogen, with a ratio of [NdV_3_]:[Al(iBu)_2_H]:[Me_2_SiCl_2_] of 1:20:3.

**Table 3 polymers-14-01406-t003:** The mean and standard deviation (mean ± SD) of the mechanical properties and crosslinking density of rubbers with the CV system.

Sample	Rubber	Tensile Strength (MPa)	Elongation at Break (%)	Moduli at 100% (MPa)	Moduli at 200% (MPa)	Moduli at 300% (MPa)	Crosslinking Density (mol/cm^3^ × 10^−3^)	Hardness (Shore A)	Compression Set (%)
1CV	PB-H	1.92 ± 0.75	674 ± 121	0.53 ± 0.09	0.76 ± 0.09	0.96 ± 0.12	9.45	54.6 ± 3.6	33.77 ± 6.36
2CV	PB-H-CB ^a^	5.92 ± 0.57	231 ± 24	2.85 ± 0.28	5.09 ± 0.64	-	29.20	80.4 ± 2.3	59.19 ± 2.52
3CV	PI-H	1.72 ± 1.17	1263 ± 25	0.17 ± 0.04	0.24 ± 0.03	0.29 ± 0.03	4.46	17.8 ± 2.3	96.68 ± 3.23
4CV	PI-H-CB ^a^	8.17 ± 0.51	419 ± 55	1.95 ± 0.13	3.83 ± 0.31	5.86 ± 0.46	24.80	73.4 ± 3.0	79.20 ± 1.71
5CV	PM-H	0.55 ± 0.10	488 ± 32	0.11 ± 0.01	0.21 ± 0.01	0.31 ± 0.03	6.09	23.2 ± 1.3	83.43 ± 11.63
6CV	PM-H-CB ^a^	2.27 ± 0.44	225 ± 65	1.14 ± 0.15	2.08 ± 0.16	-	21.2	62.6 ± 4.4	68.32 ± 2.42
7CV	PF-H	0.20 ± 0.0	181 ± 24	0.09 ± 0.01	-	-	6.44	22.4 ± 3.2	99.50 ± 0.87
8CV	PF-H-CB ^a^	2.97 ± 0.05	231 ± 13	1.16 ± 0.12	2.52 ± 0.13	-	27.30	57.2 ± 6.8	84.63 ± 1.00
9CV	PB-L	0.75 ± 0.06	663 ± 63	0.25 ± 0.00	0.35 ± 0.01	0.42 ± 0.02	3.38	36.0 ± 1.0	62.77 ± 4.05
10CV	PB-L-CB ^a^	5.15 ± 1.30	281 ± 38	1.74 ± 0.19	3.52 ± 0.60	-	21.00	72.2 ± 2.3	38.79 ± 2.42
11CV	PI-L	1.20 ± 0.14	988 ± 48	0.13 ± 0.01	0.20 ± 0.01	0.26 ± 0.01	3.76	18.0 ± 1.6	98.43 ± 1.37
12CV	PI-L-CB ^a^	8.12 ± 0.44	375 ± 20	1.94 ± 0.06	4.27 ± 0.10	6.63 ± 0.13	24.10	81.4 ± 3.4	75.14 ± 8.73
13CV	PM-L	0.20 ± 0.0	481 ± 55	0.03 ± 0.00	0.06 ± 0.00	0.10 ± 0.01	2.66	12.8 ± 1.3	97.61 ± 4.14
14CV	PM-L-CB ^a^	2.15 ± 0.44	250 ± 41	0.87 ± 0.14	1.80 ± 0.29	-	24.00	71.0 ± 2.2	75.24 ± 1.92
15CV	PF-L	-	-	-	-	-	-	-	-
16CV	PF-L-CB ^a^	0.57 ± 0.05	156 ± 24	0.37 ± 0.13	-	-	-	62 ± 5.9	69.39 ± 10.75

^a^ Carbon black-reinforced compounds.

**Table 4 polymers-14-01406-t004:** The mean and standard deviation (mean ± SD) of the mechanical properties and crosslinking density of rubbers with the EV system.

Sample	Rubber	Tensile Strength (MPa)	Elongation at Break (%)	Moduli at 100% (MPa)	Moduli at 200% (MPa)	Moduli at 300% (MPa)	Crosslinking Density (mol/cm^3^ × 10^−3^)	Hardness (Shore A)	Compression Set (%)
1EV	PB-H	1.32 ± 0.13	413 ± 48	0.59 ± 0.04	0.84 ± 0.05	1.06 ± 0.06	13.50	41.8 ± 2.9	54.88 ± 6.87
2EV	PB-H-CB ^a^	7.10 ± 0.42	313 ± 14	2.48 ± 0.10	4.36 ± 0.21	6.58 ± 0.35	29.70	69.4 ± 2.4	35.25 ± 0.57
3EV	PI-H	4.2 ± 0.54	1000 ± 0	0.35 ± 0.02	0.53 ± 0.02	0.72 ± 0.02	9.60	11.0 ± 1.2	100.00 ± 0.0
4EV	PI-H-CB ^a^	10.68 ± 0.98	400 ± 46	2.27 ± 0.22	5.03 ± 0.66	8.09 ± 0.99	29.20	68.8 ± 2.6	56.39 ± 0.37
5EV	PM-H	0.63 ± 0.05	438 ± 14	0.13 ± 0.01	0.26 ± 0.01	0.39 ± 0.01	7.17	15.8 ± 1.3	32.11 ± 6.15
6EV	PM-H-CB ^a^	6.45 ± 0.58	381 ± 47	1.39 ± 0.37	3.16 ± 0.57	5.06 ± 0.50	32.30	61.2 ± 4.1	34.47 ± 8.60
7EV	PF-H	0.20 ± 0.0	313 ± 52	0.07 ± 0.00	0.14 ± 0.01	0.21 ± 0.03	5.74	6.0 ± 1.0	68.71 ± 19.25
8EV	PF-H-CB ^a^	4.30 ± 0.30	369 ± 24	0.87 ± 0.21	1.93 ± 0.45	3.06 ± 0.72	22.30	55.60 ± 2.9	43.74 ± 3.74
9EV	PB-L	1.18 ± 0.29	825 ± 189	0.27 ± 0.08	0.42 ± 0.07	0.55 ± 0.10	6.11	53.2 ± 3.6	81.89 ± 1.20
10EV	PB-L-CB ^a^	7.98 ± 0.87	363 ± 32	1.62 ± 0.68	3.56 ± 1.67	5.83 ± 2.36	21.90	67.6 ± 2.3	37.55 ± 3.41
EV	PI-L	2.48 ± 0.10	1000 ± 0	0.19 ± 0.00	0.30 ± 0.00	0.41 ± 0.01	5.75	11.6 ± 2.7	98.14 ± 2.14
12EV	PI-L-CB ^a^	8.53 ± 0.99	431 ± 24	1.54 ± 0.19	3.63 ± 0.47	6.03 ± 0.68	17.50	75.6 ± 2.7	67.06 ± 14.99
13EV	PM-L	0.23 ± 0.05	481 ± 69	0.03 ± 0.02	0.07 ± 0.02	0.13 ± 0.04	3.50	9.8 ± 1.3	100.00 ± 0.0
14EV	PM-L-CB ^a^	2.35 ± 0.13	331 ± 24	0.62 ± 0.06	1.49 ± 0.10	2.18 ± 0.10	19.60	52.2 ± 2.2	61.47 ± 5.31
15EV	PF-L	-	-	-	-	-	-	-	-
16EV	PF-L-CB ^a^	0.95 ± 0.06	231 ± 13	0.43 ± 0.04	0.89 ± 0.02	-	-	45 ± 3.9	71.31 ± 4.60

^a^ Carbon black-reinforced compounds.

**Table 5 polymers-14-01406-t005:** Storage and loss modulus at −40 °C, T_g_, and Tan δ (maximum value) of the crosslinked materials.

Sample	Tg (°C)	Maximum Value of Tan δ	Loss Modulus at −40 °C (MPa)	Storage Modulus at −40 °C (MPa)
CV-PB-H	−78	0.97	3.58	29.61
CV-PI-H	−34	1.48	251	399.9
CV-PM-H	−42	1.43	95.63	70.17
CV-PF-H	−36	0.83	75.84	137
EV-PB-H	−79	0.26	10.99	159.7
EV-PI-H	−37	2.16	692.5	441.1
EV-PM-H	−45	0.88	62.2	88.29
EV-PF-H	−52	1.95	3.28	5.16
CV-PB-H-CB ^a^	−81	0.41	23.37	223.4
CV-PI-H-CB ^a^	−37	0.56	60.1	1353
CV-PM-H-CB ^a^	−37	0.69	405.3	513.2
CV-PF-H-CB ^a^	−42	0.45	207.6	459
EV-PB-H-CB ^a^	−84	0.23	32.84	394.1
EV-PI-H-CB ^a^	−40	0.84	1980	2420
EV-PM-H-CB ^a^	−46	0.35	219.3	1188
EV-PF-H-CB ^a^	−51	0.79	109.9	204.5

^a^ Carbon black-reinforced compounds.

## Data Availability

Not applicable.

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
