# Peer review of "Synthesis and Vulcanization of Polymyrcene and Polyfarnesene Bio-Based Rubbers: Influence of the Chemical Structure over the Vulcanization Process and Mechanical Properties"

_polymers, 2022, doi:10.3390/polym14071406_

Round 1
Reviewer 1 Report
Presented work is interesting and well prepared but should be completed in some points:
- Section 2.3.2.: Did you use extensometers during static tensile test?
- Figure 8: Figure should be enlarged. Vertical lines should be removed.
- Figure 9: Values should be supported by standard deviations
- Table 3 and Table 4: Values of mechanical properties should be supported by standard deviation
- Supplementary information: (1) Table S1 and S2 should be completed by values of ΔM and CRI (2) Please, add exemplary stress-strain curves (from static tensile test)
- In my opinion conclusions should clearly indicate the chemical structure-propoerties relationship in prepared materials
Author Response
1
Response to Reviewer 1 Comments
Manuscript ID: polymers-1644524
Title: Synthesis and Vulcanization of Polymyrcene and Polyfarnesene Bio-based
Rubbers: Influence of the Chemical Structure over the Vulcanization Process and
Mechanical Properties
Presented work is interesting and well prepared but should be completed in some points.
We would like to thank the reviewer for their positive and constructive
comments/suggestions. A detailed response to the reviewer’s comment are listed below.
Point 1: Section 2.3.2. Did you use extensometers during static tensile test?
Response 1: The measurements were manual because the extensometer was not enabled.
Point 2: Figure 8. Figure should be enlarged. Vertical lines should be removed.
Response 2: The Figure 8 was modified and enlarged according to the comment of the reviewer.
Point 3: Figure 9. Values should be supported by standard deviations.
Response 3: As suggested by the reviewer, we have modified the Figure 9 by adding the standard
deviation of the data.
Point 4: Tables 3 and Table 4: Values of mechanical properties should be supported by standard
deviation.
Response 4: As suggested by the reviewer, we have included the standard deviation of the
mechanical properties.
Point 5: Supplementary information: (1) Table S1 and S2 should be completed by values of ΔM and
CRI (2). Please, add exemplary stress-strain curves (from static tensile test).
Response 5: The values of ΔM and CRI were included in the supplementary information according
to the comment of the reviewer. In addition, some examples of stress-strain curves were included.
Point 6: In my opinion conclusions should clearly indicate the chemical structure-properties
relationship in prepared materials.
Response 6: We consider that the conclusion generally covers the most imporant points with respect
to the chemical structure of the crosslinked materiasl and its effect on the properties. However, we
made some slight changes.
Reviewer 2 Report
The manuscript (polymers-1644524-peer-review-v1) dealing in structure-property-performance evaluation of bio-based rubbers seems to be significant, interesting, and timely with promising application prospects. Authors have mainly concentrated in sulphur-vulcanization, comparative crosslinking status, and associated mechanical properties among various non-reinforced/ reinforced general purpose rubber composites composed of either conventional petroleum-derived synthetic rubbers, i.e., polybutadiene (PB) and polyisoprene (PI), or these newly reported bio-based rubbers, i.e., polymyrcene (PM) and polyfarnesene (PF). However, the present form of the manuscript still needs some improvements based on the underlying points:
- In section 2.2.2, the authors have not mentioned detailed parameters, i.e. nip gap, sequence of adding ingredients etc., at the time of two-roll mixing. Moreover, the purpose behind addition of each ingredient should be mentioned in brief.
- At the time of mixing, authors have not added any antioxidant or any processing aid like calcium stearate. Why is it so?
- In the rheometric curves of conventionally vulcanized rubbers (Fig. 2), the final torque value becomes almost stationery with respect to the time, i.e., no reversion has been noticed. In contrast, in the rheometric curves of efficiently vulcanized rubbers (Fig. 3), especially the reinforced samples, show clear presence of reversion, i.e., the torque value has been deteriorated with respect to the time. Since, the reversion phenomenon is related to the breakdown of polysulphide links, these reinforced EV samples should contain frequent polysulfides. However, due to lesser availability of added sulphur, polysulfide formation in EV rubbers is less probable than that in CV rubbers. As a whole, the rheometric results contradict the theory, and I suggest reinvestigation of the overall results. In this regard, I suggest to go through the following papers: Korean Journal of Chemical Engineering 34 (2017)1416–1434; European Polymer Journal 81 (2016) 98-113; Korean Journal of Chemical Engineering 35 (2018) 1889–1910 .
- While reporting and comparing the mechanical properties (Table 3 and 4), inclusion of moduli at 100%, 200%, and 300% elongations are highly important and well-connected to the crosslinking density. Accordingly, authors should incorporate and correlate moduli values with the crosslinking densities. Moreover, in each sample, error bar should be provided.
- In the second para of section 3.5, there is a serious unwanted error while expressing the Tan d. This has been mistakenly represented as the ratio of storage to loss moduli.
- Additionally, I suggest characterization of the as-prepared samples by hysteresis (for evaluating the heat-build up) and thermogravimetric analyses to evaluate the comparative thermal properties of the compositions. In fact, the nature of vulcanization and crosslinking directly influence the thermal stability of rubber backbone.
Altogether, the paper needs some essential modifications. Otherwise, in the present form, the manuscript is not recommended for further processing.
Author Response
1
Response to Reviewer 2 Comments
Manuscript ID: polymers-1644524
Title: Synthesis and Vulcanization of Polymyrcene and Polyfarnesene Bio-based
Rubbers: Influence of the Chemical Structure over the Vulcanization Process and
Mechanical Properties
The manuscript (polymers-1644524-peer-review-v1) dealing in structure-property-performance
evaluation of bio-based rubbers seems to be significant, interesting, and timely with promising
application prospects. Authors have mainly concentrated in sulphur-vulcanization, comparative
crosslinking status, and associated mechanical properties among various non-reinforced/ reinforced
general purpose rubber composites composed of either conventional petroleum-derived synthetic
rubbers, i.e., polybutadiene (PB) and polyisoprene (PI), or these newly reported bio-based rubbers,
i.e., polymyrcene (PM) and polyfarnesene (PF). However, the present form of the manuscript still
needs some improvements based on the underlying points.
We would like to thank the reviewer for their positive and constructive
comments/suggestions. A detailed response to the reviewer’s comment are listed below.
Point 1: In section 2.2.2., the authors have not mentioned detailed parameters, i.e. nip gap, sequence
of adding ingredients etc., at the time of two-roll mixing. Moreover, the purpose behind addition of
each ingredient should be mentioned in brief.
Response 1: Following the reviewer’s suggestion, we have included more details in the description
of the rubber compounding.
Point 2: At the time of mixing, authors have not added any antioxidant or any processing aid like
calcium stearate. Why is it so?
Response 2: Our work presents conventional and efficient systems for vulcanization of elastomers
using a basic recipe for rubber based on the paper Rubber Chemistry and Technology 38 (1965) 1-14,
where no antioxidant is used, which has also been followed in other works such as Iranian Polymer
Journal 24 (2015) 289-297, KGK Kautschuk Gummi Kunststoffe 58 (2005) 638-643 and Rubber
Chemistry and Technology 87 (2014) 21-30.
Point 3: In the rheometric curves of conventionally vulcanized rubbers (Fig. 2), the final torque
value becomes almost stationery with respect to the time, i.e., no reversion has been noticed. In
contrast, in the rheometric curves of efficiently vulcanized rubbers (Fig, 3), especially the reinforced
samples, show clear presence of reversion, i.e., the torque value has been deteriorated with respect
to the time. Since, the reversion phenomenon is related to the breakdown of polysulphide links,
these reinforced EV samples should contain frequent polysulfide formation in EV rubbers is less
2
probable than that in CV rubbers. As a whole, the rheometric results contradict the theory, and I
suggest reinvestigation of the overall results. In this regard, I suggest to go through the following
papers: Korean Journal of Chemical Engineering 34 (2017) 1416-1434; European Polymer Journal 81
(2016) 98-113; Korean Journal of Chemical Engineering 35 (2018) 1889-1910.
Response 3: Certainly, as mentioned by the reviewer, the EV vulcanized rubbers show a slight
reversion in the rheometric curves which we had overlooked. The theory found in various
publications indicates that this reversion phenomenon commonly occurs in conventional
vulcanization. However, the following papers KGK Kautschuk Gummi Kunststoffe 58 (2005)
638-643, Polymer Testing 88 (2020) 106524 present in their results a level of reversion in the
rheometric curves of efficient vulcanization attributing it to the conversion of some disulfide
crosslinks to monosulfides, decreasing the crosslinking bridges. In addition, in our case, the absence
of antioxidant could favor the reversion, according to Pat et al., Polym Eng Sci. 88 (2021) 2616, the
ozone and oxygen are two factors that are responsible for breaking polymer chains with a
simultaneous reduction of molecular weight. In this sense, the presence and the selection of proper
antioxidant restrict such effects. In our case in conventional vulcanization there is no explanation
for the non-reversion so far.
Point 4: While reporting and comparing the mechanical properties (Table 3 and 4), inclusion of
moduli at 100%, 200%, and 300% elongations are highly important and well-connected to the
crosslinking density. Accordingly, authors should incorporate and correlate moduli values with the
crosslinking densities. Moreover, in each sample, error bar should be provided.
Response 4: As suggested by the reviewer, we have included the mean values of moduli at 100%,
200%, and 300% and its standard deviation in Tables 3 and 4 with a brief discussion of the results.
Point 5: In the second part of the section 3.5, there is a serious unwanted error while expressing the
Tan . This has been mistakenly represented as the ratio of storage to loss moduli.
Response 5: This error has been corrected in the manuscript, thank you for pointing it out.
Point 6: Additionally, I suggest characterization of the as-prepared samples by hysteresis (for
evaluating the heat-build up) and thermogravimetric analyses to evaluate the compartive thermal
properties of the compositions. In fact, the nature of vulcanization and crosslinking directly
influence the thermal stability of rubber backbone.
Response 6: Unfortunately, due to lack of material, it is not possible for us to do the
characterization that suggests. However, we consider that the results shown are sufficient to make a
good discussion.
Round 2
Reviewer 2 Report
The authors have considered some of the issues raised in the earlier review. However, this revised article may be considered for publication.